# How Administrative Regulation Institutional Factors Affect the Business Efficiency in a Region: A Case Study of Russian Regions

**Leyla Gamidullaeva** [1,*] and **Saniyat Agamagomedova** [2]

1   Department of Marketing, Commerce and Service, Penza State University, 40, Krasnaya Str., 440026 Penza, Russia
2   Institute of State and Law of the Russian Academy of Sciences, 119019 Moscow, Russia; saniyat_aga@mail.ru
*   Correspondence: gamidullaeva@gmail.com

**Abstract:** Different business environments may have differential impacts. The improvement in the institutional environment and increased certainty about the future greatly impact entrepreneurial activity and business results. The reform of inspection and regulatory enforcement activities aims at improving the business environment, i.e., property, lawful rights, and interests against a variety of risks. Although the reform has been ongoing for several years in Russia, no systematic, independent evaluation of its efficiency and impact on the business efficiency at the meso-level of the Russian economic system has yet been implemented. The research aimed to investigate administrative barriers to business in (1) subdivisions of the Russian Federation on the example of monitoring and supervision activities of state bodies, and (2) the provision of public services. Besides, we measured the impact of administrative regulation on the effectiveness of entrepreneurial activity in the region. Findings from this research are useful in managerial practice, and for academic researchers and policymakers in emerging economies to adopt and consider to improve further the contribution of the administrative regulation institutions in regional economic development.

**Keywords:** administrative regulation; inspection; business efficiency; region

## 1. Introduction

The literature concerning the research problem linked to this paper highlights the significant role played by common institutions and policies when affecting a firm's economic results. Different business environments may have differential impacts. The improvement in the institutional environment and increased certainty about the future have much impact on entrepreneurial behavior (Gamidullaeva et al. 2020). The concept of a barrier is used quite actively in the scientific literature to position diverse obstacles in the development of certain institutions, processes, and activities, wherein the 'barrier' category is most frequently used in relation to the business environment and the sphere of entrepreneurship. The latter is based on the principles of a certain freedom of economic activity of an economic entity, limited by the criteria of security and economic feasibility, clearly regulated by the current legislation. Such varieties of barriers as barriers to entry (Ross 2020; Hoshi et al. 2003; Narovlyanskaya and Kartasheva 2012) and trade barriers (Yeats 1979) are sufficiently studied in the current scientific literature. Besides, there are three primary groups of barriers for every business. They include (1) capital (financial backing), (2) knowledge (level of education and/or experience), and (3) persistence (staying power) (Zhang et al. 2017).

At the end of the last century, scientists stated the fact that the institutional framework has a significant impact on the efficiency and rate of economic growth. They recognized that politically open societies that support the rule of law grow faster and are much more efficient than societies in which certain freedoms are restricted by the regulator (Scully 1998). When studying the impact of regulation on economic and social processes, scientists rightly

identify two distinct trends in regulation over the last two decades of the last century: (1) an unprecedented increase in the number of new requirements in the field of health, safety, and the environment, and (2) a significant economic deregulation (Guasch and Hahn 1997; Hahn 1998).

In modern science, there is a study of the relationship between the quality of the administrative environment for business, and the propensity to become an entrepreneur in the EU member states against the background of other world economies: BRIC, Japan, and the USA. The purpose of such studies is to determine the impact of regulation on the difference in GDP between developed and underdeveloped countries (Anders 2011).

Currently, the naturally managerial concept of 'administrative barriers' is actively used in Russian legal and economic science. Having appeared relatively recently, this concept was first used in the Decree of the President of the Russian Federation On Measures for Eliminating Administrative Barriers to Development of Entrepreneurship (No. 730 of 29 June 1998) (Presidential Executive Office 1998). Initially, the considered 44 concept was exclusively used when implementing the entrepreneurial activity. Moreover, 45 scientists interpreted it as "obstacles arising from the organization and implementation of 46 entrepreneurial activity in small businesses created by individual officials of the executive 47 branch" (Filimonov 2001). Later, the concept of excessive administrative procedures (actions) began to be used in the normative array, which, in our opinion, coincides with the concept of an administrative barrier and is an expression of the latter in the context of the administrative procedural approach. A procedural approach to the concept of an administrative barrier investigates the problem of excessive public services. S. E. Naryshkin and T. Ya. Khabrieva rightly spoke of fictitious and excessive public services; the latter are administrative barriers, in their opinion. Excessive public service (administrative barrier) is a service exceeding an entity's private introduction costs over its private and public benefits, considering the income effect (Naryshkin and Khabrieva 2006).

The scientists have established a key criterion for classifying the administrative procedure as excessive, and acting as an administrative barrier. It is the final result of the administrative procedure (in terms of private and public interests) in relation to the costs (efforts) to achieve it. Thus, barriers in business are of a broader concept; these barriers include barriers of economic (financial), organizational and legal (administrative barriers), and other natures. Administrative barriers to business exist and develop at the junction of economic and legal institutions; they are associated with such universal categories as business performance. "Administrative formalities take an infinite variety of forms, and it is impossible to develop a generic definition. To add to the confusion, the terms administrative formalities, administrative barriers, and administrative procedures are used interchangeably" (Jacobs and Coolidge 2006; Kazachenko and Samoilenko 2007).

Our scientific position is that the regulatory framework or administrative regulation is present as an indispensable element in any economic system. Despite the fact that scientists call them in different terms (administrative procedures, administrative formalities, administrative barriers, regulatory framework), we believe that their meaning lies in the fact that public authorities restrain the development of entrepreneurial and other economic activity, freedom based on security requirements, and other purposes of a public nature. The private interests of business can be realized only considering the public interests (interests of the majority), which the state is also called upon to pursue through the establishment of mandatory requirements for the economic activity of private entities. In this regard, any mandatory procedures established by law for the interaction of the state with economic entities can be viewed from the most general positions as barriers and obstacles limiting their freedom and independence.

Moreover, administrative formalities or barriers can be present in all forms of interaction between business and public authorities: the provision of public services, the licensing system of the state, state control, or state supervision (in some legal acts, the latter are used as synonyms, in others, they are delimited). Thus, we conclude that initially, the concept of administrative barriers in Russian practice is used in a narrow sectional sense

(in entrepreneurial activity, in the framework of competition development, etc.). Later, the problem of excessive administrative barriers and state regulation gained a wider understanding and became the basis for a comprehensive intersectional direction of state policy. It was noted that administrative barriers are a common obstacle to the growth of efficiency of state and public institutions (Vasiliev et al. 2016).

Later, the state policy to reduce administrative barriers was also implemented as part of the Competition Development Program (Government of the Russian Federation 2021).

It is interesting that this program contained certain types of administrative barriers classified in terms of (1) the functioning sphere (domestic and foreign trade barriers, barriers between local markets); (2) the period (stage) of the relevant activities (administrative barriers to entry into the market and administrative barriers to business) in terms of content (economy, information); (3) the sector (sub-sector) of the national economy (administrative barriers in carrying out exploration, production, storage and transportation of oil; administrative barriers at the stage of land allocation; administrative barriers in the design and construction); and (4) the degree and nature of the obstructing effect (significant, prohibitively high, unreasonable, unjustified, artificial, insuperable). In the 2000s, several theses on the economic aspects of administrative barriers both in the economy (Zakharov 2004) and entrepreneurship (Ushakov 2005; Voloshina 2006; Sosunov 2007) were defended.

In addition, sociologists considered information and communication barriers in public administration (Khizhnyakov 2011).

Since part of state monitoring and supervision activities reform is carried out in Russia, the interest of scientists and practitioners concerning the problems of administrative barriers is growing. The scientists study individual aspects of the existence of administrative barriers in various fields (Stepanenko 2018), compare them with other types of barriers (Zubarev 2018), and examine them in the context of different types of state control (Agamagomedova 2016, 2017). A survey of Rostov entrepreneurs (2164 respondents) showed that the presence of administrative barriers is a negative factor hampering the effective development of small businesses; it is in third place after such factors as a high level of taxation and an imperfect regulatory framework (Sosunov 2007).

Current studies of administrative barriers are often localized in particular countries or even individual sectors of the national economy. For example, some authors explore the transformation of governance in agriculture in Vietnam, when the elimination of price controls and public procurement systems increased the efficiency of farmers, and when collaborative governance (CG) and inter-organizational governance (IOG) became the key to successful reforming regulatory and management mechanisms (Dung et al. 2023).

The current research aim was to investigate administrative barriers to business development in regions of the Russian Federation on the example of monitoring and supervision activities of state bodies and the provision of public services. Next, it was necessary to assess the impact of administrative regulation on the effectiveness of entrepreneurial activity (business efficiency) in a region to identify the available reserves in this area. It is the reserve that regional entrepreneurs have in resisting administrative pressure from the authorities, a reserve that allows them not only to comply with the mandatory requirements of the state, but also to develop further.

To summarize the direction of this research, the following two hypotheses framed the direction of this study:

- *Hypothesis I*: The level of administrative control has a significant positive effect on the net financial result of organizations in the regions of the Russian Federation;
- *Hypothesis II:* There is a significant non-monotonic dependence of the share of profitable enterprises on the level of monitoring and supervision regulation.

The proposed study is intended to determine the level of redundancy of administrative procedures for the interaction of regional business with the authorities on the example of control and supervisory interaction. The relevance of studying precisely these aspects is connected with both the ongoing reform of state control and supervision in Russia, and the transformation of the Russian economy in the face of increased sanctions pressure.

The next section discusses the recent literature on administrative burdens measuring and reduction to acknowledge the previous research on the issue related to the subject discussed in this paper. This paper presents a research methodology, which includes the process of selecting variables. The paper then presents the data set used and the results of the empirical analysis. The contribution of this paper is the development and testing of empirical methodological approach to measuring the effects of administrative regulation on business effectiveness in Russia. Findings from this study are useful in managerial practice, academic researchers, and policymakers in emerging economies to adopt and consider to improve (and increase) the contribution of administrative regulation institutions in regional economic development.

## 2. Literature Review

The quality of administrative institutions is considered a crucial element of any institutional environment; the more effective administrative institutions, the lower administrative barriers in the economy.

Several fundamental approaches should be highlighted when considering administrative barriers in the economy.

*The first approach* reflects the institutional concept (Auzan and Kryuchkova 2001, 2002; Malikov 2003; Nikolaev and Shulga 2003). In this context, administrative barriers are formal mandatory rules for conducting business in the markets for goods and services established by state authorities and local self-government when the private introduction costs of an economic entity exceed its private and public benefits, taking into account the income effect (Auzan and Kryuchkova 2002).

*The second scientific position* considers administrative barriers in relation to integration processes, which are embodied in the acts of the Eurasian Economic Union (Balandina 2011).

*The third approach* is related to the category of procedures, and is based on positioning the administrative barrier as an unnecessary administrative procedure (or its part). Here, administrative barriers are considered concerning the procedures for exercising public functions and providing public services (Rodrik et al. 2004; Naryshkin and Khabrieva 2006; Madsen 2009; Mason and Brown 2011; Agamagomedova 2017).

In the framework of the procedural approach, an administrative barrier is an excessive administrative procedure or action in a universal form.

Administrative barriers to business are often considered in the context of corruption, abuse, and discretion of authorities. The acts of state power contain provisions according to which the causes and conditions that create corruption and create administrative barriers are combined (Presidential Executive Office 2005).

As for the correlation of administrative barriers in the economy and business, it should be noted that in most cases, these concepts have a common nature and coincide. We believe that administrative barriers in the economy include all kinds of formal requirements for doing business, which ultimately leads to cost overruns for entrepreneurs. At the same time, the category of administrative barriers in the economy may include other obstacles to the development of the economy that are not directly related to profit-making by business entities. For example, we can talk about the institution of self-regulation, and obstacles to its development can be interpreted as an administrative barrier.

In turn, administrative barriers to business can be divided into several types: barriers to (1) the provision of public services; (2) obtaining a certain administrative and legal status (e.g., obtaining the status of a small or medium business); (3) passing state monitoring; and (4) supervision. The latter are directly related to the business—the sphere controlled by the state to ensure security, the interests of consumers of goods and services, and other public interests.

State control (supervision) is understood as the activities of state bodies aimed at preventing, detecting, and suppressing violations of mandatory requirements, carried out through preventive measures.

There are several types of administrative barriers for businesses when passing state monitoring and supervision.

*First*, they can be divided into legal and illegal ones.

Legal administrative barriers are also called formal barriers. These are restrictions on entrepreneurial activity that are established in legislative acts to prevent unscrupulous entities from entering the market (Tereshchenko and Kalmykova 2011).

Illegal administrative barriers arise in cases of a gap in the law and the exercise of discretion by a public representative.

*Second*, administrative barriers can be systematized by the nature of costs:

- Time costs;
- Financial (material) costs;
- Costs associated with the number and types of documents requested by the regulatory body;
- Costs related to organizational barriers to continued business growth

*Third*, administrative barriers to business can be divided, namely:

- Barriers arising when a company enters a certain market of goods (services);
- Barriers arising from the functioning of the company and its main activities;
- Barriers arising when a company develops new types of economic activity (e.g., foreign economic activity), new categories of goods (services), or when a company enters new markets, etc.

*Fourth*, administrative barriers to business depend on the category of business and (or) the scope of its activities. In the hospitality industry, administrative barriers will have their own content and structure, and they will be somewhat different in the tourism industry.

*Fifth*, in passing state monitoring and supervision, administrative barriers to business can be divided according to the level of such control: monitoring and supervision by federal executive bodies, and municipal monitoring and supervision.

The effects of administrative barriers on business can be divided into purely economic (narrow approach) and social (broad approach).

In this paper, we consider the barriers associated with the administrative costs of entrepreneurs in the process of current economic activity.

An actual basis for direct losses for the population from rising retail prices are official, and shadow payments that business entities are forced to make to formally fulfill the requirements that constitute administrative barriers to doing business. Due to their economic nature, these payments represent various transaction costs that are the reason for the low business activity of economic agents (Gamidullaeva and Tolstykh 2017; Vasin et al. 2018; Gamidullaeva 2019b).

A great amount of research on the problem of assessing and reducing administrative barriers to business presented in foreign literature is devoted to a case study, an overview of best regulatory practices, and administrative support for business in the EU and other countries of the world; these are works by R. E. Berney (1980), Berney and Swanson (1982), Guasch and Hahn (1997), Scully (1998), Spolaore and Wacziarg (2014), R. W. Hahn (1998, 1999), Evans and Walpole (2000), Hahn and Litan (2000), Gree and Thurnik (2003), A. Gibb (2000), F. Sterzel (2001), Chittenden et al. (2002), M. Munnich (2004), etc. An extensive review of the world's best regulatory practices to reduce the burden of business (administrative burden) is presented by the European Commission (2012).

Boeheim et al. (2006) analyzed four case studies using EU countries as an example. As a result, the authors developed several approaches for assessing administrative barriers to business, including the Standard Cost Model, and also conducted some research in the context of economic sectors. R. Arendsen et al. (2014) attempted to assess the reduction in the negative impact of administrative barriers through the introduction of e-government in the Netherlands.

The specificity of the research on administrative regulation and its impact on businesses in various countries is mostly associated with the digitalization of interaction be-

tween the state and economic entities, with special (targeted) measures to support businesses at the national and regional levels. At the same time, for countries with transitional economies, the approach of considering the regulation system itself as a barrier factor for entrepreneurship development is more in demand. In contrast, the most visible element of this system is state control and supervision. State control and supervision are forms of interaction between business and public authorities, the obligation and scope of which most significantly affect the development of entrepreneurial and other economic relations, including at the regional level. Scientists who view administrative barriers as an institution of a transformational economy also include state control and supervision (Narovlyanskaya and Kartasheva 2012).

Current research in the field of administrative barriers also focuses on the specifics of administrative and legal regulation in the context of quarantine measures related to countering the COVID-19 pandemic. We highlight the administrative features of protecting the rights of economic entities under quarantine conditions, and emphasize the role of the state in ensuring business guarantees (Slobodianiuk et al. 2021).

Scientists are also actively exploring the factors of administrative reforms in various countries, taking into account the specifics of a national government. For example, F. C. Liao examines administrative reform and the conditions for reducing administrative barriers in China, and concludes that administrative reform can optimize the business environment in three areas: balanced development path, reform path, and resource path. They understand the reduction in the administrative burden on the business as the basis for promoting the business environment and stimulating its development (Obeng 2019; Liao 2020).

Research of administrative regulation in the business sphere is naturally differentiated depending on (1) the category of business (small and medium business, other classifications) and (2) the category of the country in which regulation is carried out (developed countries, countries with economies in transition, etc.). Researchers also consider the management aspects of small business development in developed countries and transitional economies (for example, Ukraine), provide their comparative analysis (Zahorskyi et al. 2019). With regard to individual developing countries (for example, Peru), scientists talk about public-private partnerships and contractual regulation that minimized the risks associated with administrative interference by governments in private investment (Fariza 2012; Diaz 2017).

Concerning certain areas of government regulation, scientists emphasize the importance of voluntary programs that combine elements of public and private governance. An example is the reduction of the administrative burden that agricultural producers face when wanting to obtain organic certification (Carter et al. 2018). At the same time, in particular countries and areas of government regulation, scientists note an increase in the repressiveness of administrative pressure and the practice of sanctions (or penalties) regulatory measures (Tollenaar 2018).

Of particular value are studies containing a quantitative assessment of the administrative impact formed by regulatory legal acts containing mandatory requirements, including ones in the security field. At the same time, an important issue is finding the optimal balance between reducing the administrative burden on business and preventing negative impact on security, as well as recommendations for eliminating and simplifying administrative requirements and procedures based on the calculation of costs for each administrative requirement (Pilvere et al. 2013; Rauch et al. 2013; Prykhodko 2015).

It is noteworthy to point out that there is a lack of quantitative and qualitative studies explaining and characterizing the institutional environment (Gamidullaeva et al. 2020). There have been practically no such large-scale studies devoted to the development of methodological approaches, and the assessment of the impact of administrative barriers on the effectiveness of entrepreneurial activity in Russia as a transitional economy. This research fills this gap; we attempt to assess the impact of administrative barriers on entrepreneurship development based on the example of Russian regions.

### 3. Methodology

The data sources for the research included:

- Reports of the Ministry of Economic Development of Russia On the Implementation of State Monitoring (Supervision), Municipal Control in Relevant Areas of Activity, and the Effectiveness of Such Monitoring (Supervision) for the period 2015–2018 (Federal Customs Service 2018);
- Report of the Russian Union of Industrialists and Entrepreneurs in 2018;
- Reports of federal executive authorities and constituent entities of the Russian Federation on the results of monitoring and supervision activities (Presidium of the Presidential Council 2016);
- Results of a sociological survey of entrepreneurs in the Volga Federal District, conducted in 2016 and 2019 (1090 respondents involved);
- Scientific publications on state monitoring and supervision, administrative barriers to business when passing state monitoring and supervision.

The conducted research logic is presented in Figure 1.

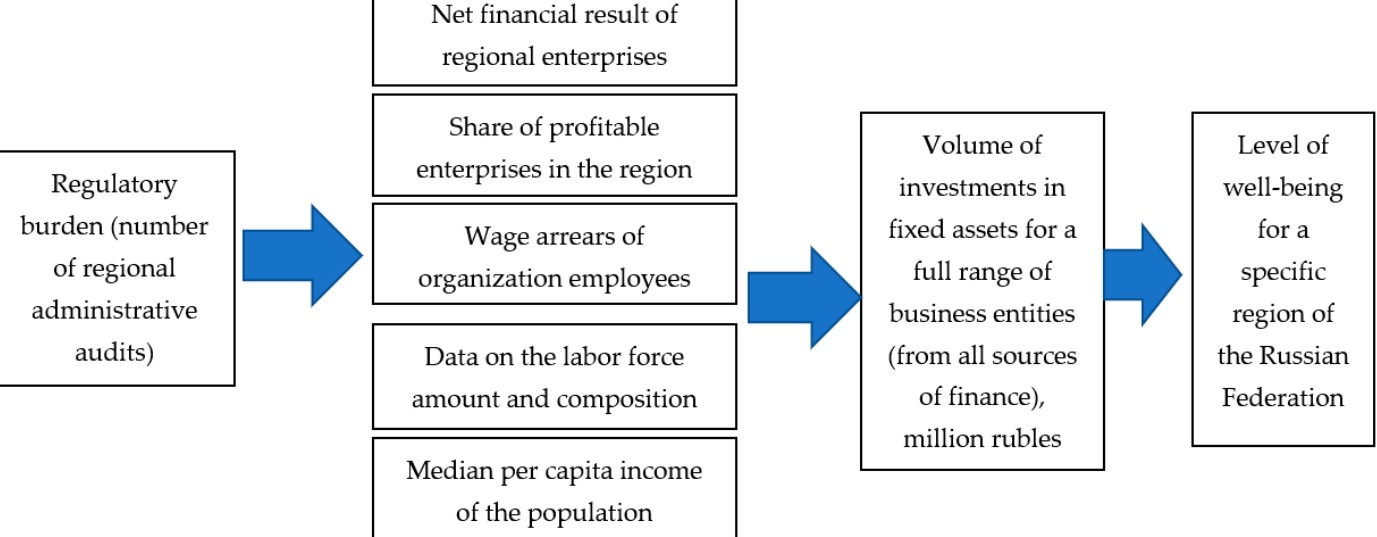

**Figure 1.** Regulations, investment, and business development.

Reports and records of state bodies and public organizations are publicly available on the official websites of these departments. A sociological survey of entrepreneurs in the Volga Federal District was carried out on the basis of the Customs Service Center at Penza State University, which implements advanced training programs for business representatives in the region. The survey was conducted in two stages, in 2016 and 2019. The survey results were obtained based on processing questionnaires completed by entrepreneurs in electronic and written form.

We worked with panel data since they allow us to take into account both the spatial and temporal components of the dependence under consideration. Panel data make it possible to control for variables that are unobservable and for variables that change over time but differ across regions. Such data allow us to more accurately take into account the individual heterogeneity of the sample.

The data used have determined the research method. Random and fixed effects models, which are widely used in the scientific literature, are employed as the main method.

It is necessary to use methods for panel data analysis to evaluate the effectiveness of monitoring and supervision measures in the Russian regions. There are two generally accepted methods for working with panel data that allow estimating model coefficients (Anatolyev 2003):

- Fixed effects model (FE);
- Random effects model (RE).

  Dependent variables:
  Model 1

- PR—net financial result of organizations (excluding small businesses), million rubles;

  Model 2

- PR_share—the share of profitable enterprises and organizations (excluding small businesses), %;

  Independent variables:

- Income—median per capita income of the population (data for the II–III quarter), rubles;
- Debt—wage arrears of organization employees (as for December 1), thousand rubles;
- Data on the amount and composition of the labor force aged 15–72 years:
  ○　emp—employed, thousand people;
  ○　U—unemployment rate, %;
- Invest—volume of investments in fixed assets for a full range of business entities (from all sources of finance), million rubles.

If we find evidence that differences between regions affect the dependent variable, then we will use the RE model. The Hausman test (Hausman 1978) is used to make an informed choice between FE and RE.

Since both methods are consistent, in the absence of correlation, we should evaluate the difference between them and choose RE if this difference is relatively small (RE is more effective in this case).

Having described the main technical details of the methodology, let us calculate the results.

### 3.1. Study Rationale

From a theoretical point of view, the influence of control actions on business activities is described by F. Blanc (2012), who examined examples of the implementation of regulatory authorities' reforms in Europe. The author identified the following main channels of influence:

- Time and financial expenses for verification preparation, direct verification, and subsequent control;
- Layered and grey areas of the specific inspection structure;
- Impact on investment decisions and expectations for potential growth.

Solodilova et al. (2016) noted a significant impact of the existing parameters of administrative regulation of entrepreneurial activity in the formats of administrative assistance and pressure on the investment climate of the region. This fact justifies the presence among the variables of the investment model in fixed assets. In addition, the authors analyzed the number of formal costs of business entities for the implementation of administrative requirements, norms, and rules in ten areas of state regulation, including tourism, consumer protection, state and municipal procurement, etc.

According to the analysis results (Solodilova et al. 2016), it was established that the main costs of entrepreneurs relate to the sphere of labor and employment (363.2 billion rubles), which is the basis for including model variables related to the labor force amount and age composition, and wage arrears of organization employees.

Based on a survey of private business entities (mainly small businesses, including individual entrepreneurs) in 28 Russian regions, Dobrolyubova and Yuzhakov (2017) have concluded that a significant level of informal payments occurs during audits, which directly leads to a decrease in business profitability and lower profits. Thus, the inclusion of model

variables characterizing the amount of profit and the share of profitable organizations seems necessary.

In addition, this research focused on the influence of the level of monitoring and supervision regulation on the general welfare of residents of a particular region of the Russian Federation. In this regard, a hypothesis was put forward about the non-monotonic effect of the number of audits on the average per capita income of the population.

### 3.2. Sampling Design

The data of the Ministry of Economic Development of Russia (2020) on the number of audits conducted by executive authorities of the federal subjects of the Russian Federation in the period from 2016 to 2018 were used as a basic variable characterizing the control and supervisory component (CONTROL variable).

### 3.3. Data Collection and Statistical Tools for Analysis

Current data from the Federal State Statistics Service were used to determine the efficacy of monitoring and supervision activities. The list of variables used is presented below:

- PR—net financial result of organizations (excluding small businesses), million rubles;
- PR_share—the share of profitable enterprises and organizations (excluding small businesses), %;
- Income—median per capita income of the population (data for the II–III quarter), rubles;
- Debt—wage arrears of organization employees (as for 1 December), thousand rubles;
- Data on the amount and composition of the labor force aged 15–72 years;
- Emp—employed, thousand people;
- U—unemployment rate, %;
- Invest—volume of investments in fixed assets for a full range of business entities (from all sources of finance), million rubles.

STATA statistical software package was used for empirical analysis and calculations. In addition, MS Excel was used for intermediate calculations.

The net financial result of organizations (excluding small businesses) in million rubles is implied. The net financial result (the amount of accounting profit a company has left over after paying off all its expenses) is the final financial result identified on the basis of the accounting records of all business operations of organizations. It represents the amount of profit (loss) from the sale of goods, products (works, services), fixed assets, and other property of organizations and income from non-sales operations, reduced by the amount of expenses on these operations. Non-sales income and expenses area fines, penalties, and forfeits for violation of the terms of contracts, as well as profit (loss) of previous years, revealed in the reporting year, exchange differences, etc. Data on the net financial result of organizations are given in actual prices, structure, and methodology of the relevant years.

### 4. Results

At the first stage of the empirical study, it is supposed to evaluate the influence model for the number of administrative audits on the total profit of regional enterprises:

$$PR_{it} = const + \beta * control_{it} + \acute{\varepsilon}_{it}, \tag{1}$$

where:

- $const$ $\beta$ are the parameters to be evaluated by the least squares method in the course of further analysis;
- $\acute{\varepsilon}_{it}$ is a stochastic component of the model. The specific nature of the error will be determined later in the paper.

The results of the model (1) evaluation using RE and FE methods are presented in Table 1.

**Table 1.** Model (1) evaluation results.

|  | (RE) | (FE) | (RE) | (FE) |
|---|---|---|---|---|
| VARIABLES | pr | pr | lnpr | lnpr |
| control | 9.235 ** | −2.528 |  |  |
|  | (4.617) | (4.984) |  |  |
| lncontrol |  |  | 0.211 ** | −0.0388 |
|  |  |  | (0.0843) | (0.0893) |
| Constant | 101,160 ** | 150,660 *** | 8.941 *** | 11.04 *** |
|  | (43,711) | (21,875) | (0.691) | (0.714) |
| Observations | 255 | 255 | 226 | 226 |
| R-squared |  | 0.002 |  | 0.001 |
| Number of region | 85 | 85 | 79 | 79 |

Standard errors in parentheses *** $p < 0.01$, ** $p < 0.05$.

As can be seen from the table, the level of administrative control has a significant (based on the Hausman test, the random effects model is preferable to the fixed effects model) positive effect on the net financial result of organizations in the regions of the Russian Federation.

To verify the results, we estimated a logarithmic model, reducing the effect of significant outliers in the data (e.g., Moscow and St. Petersburg). The evaluation of the logarithmic model has confirmed the previously obtained results.

Further, we considered the influence of the number of administrative audits on the share of profitable companies in the region. Within the basic hypothesis framework, a non-monotonic share dependence of profitable enterprises on the level of monitoring and supervision regulation was assumed. The following model was evaluated:

$$PR\_share_{it} = const + \beta * control_{it} + Y * control_{it} + \acute{\varepsilon}_{it} \qquad (2)$$

Table 2 presents the evaluation results of this model.

**Table 2.** Model (2) evaluation results.

|  | (RE) | (FE) | (RE) | (FE) |
|---|---|---|---|---|
| VARIABLES | pr_share | pr_share | pr_share | pr_share |
| control |  |  | 0.00142 *** | 0.00121 * |
|  |  |  | (0.000428) | (0.000664) |
| sq_CONTROL |  |  | −(4.71 × 10⁻⁸) * | −4.30 × 10⁻⁸ |
|  |  |  | ($2.45 \times 10^{-8}$) | ($3.40 \times 10^{-8}$) |
| lncontrol | 3.191 *** | 2.765 ** |  |  |
|  | (0.638) | (1.191) |  |  |
| Constant | 44.41 *** | 47.77 *** | 65.23 *** | 65.95 *** |
|  | (5.077) | (9.392) | (1.309) | (1.905) |
| Observations | 255 | 255 | 255 | 255 |
| R-squared |  | 0.031 |  | 0.022 |
| Number of region | 85 | 85 | 85 | 85 |

Standard errors in parentheses *** $p < 0.01$, ** $p < 0.05$, * $p < 0.1$.

As can be seen from the table, the hypothesis is confirmed by real data. In the absence of a proper level of control, an increase in monitoring and supervisory actions leads to an

increase in the number of profitable enterprises. However, upon reaching a certain level, further growth of regulation has a significant negative effect on the activities of enterprises in terms of their profitability.

According to Solodilova et al. (2016), the level of administrative control significantly impacts the level of investment activity in the region. We suggested the presence of a non-monotonic dependence in this case.

The model evaluation results using the least squares method are presented in Table 3.

**Table 3.** Dependence model evaluation results between the level of investment activity and the number of audits in the region.

|  | (1) | (2) | (3) | (4) |
|---|---|---|---|---|
| VARIABLES | invest | invest | invest | invest |
| control | −3.618 | −6.829 *** | 19.69 *** | 18.09 *** |
|  | (2.504) | (2.492) | (4.703) | (4.538) |
| sq_CONTROL |  |  | −0.00140 *** | −0.00147 *** |
|  |  |  | (0.000242) | (0.000232) |
| Constant | 227,673 *** | 241,186 *** | 178,662 *** | 187,517 *** |
|  | (38,494) | (10,936) | (39,053) | (13,013) |
| Observations | 255 | 255 | 255 | 255 |
| R-squared |  | 0.043 |  | 0.226 |
| Number of region | 85 | 85 | 85 | 85 |

Standard errors in parentheses *** $p < 0.01$.

Based on the results of the calculations, we concluded that, with an insufficient level of control, the investment climate in the region is at a non-optimal level. However, upon reaching the optimal level, further tightening of administrative barriers adversely affects the level of investment in the region.

The optimal level of the investment climate is understood as the level that ensures the maximum share of profitable companies in the region.

The next stage of the study is the analysis of the influence of monitoring and supervision functions on labor market indicators (number of employees, unemployment rate, and level of wage arrears to employees of enterprises in the region) (Table 4).

**Table 4.** Dependence model evaluation results between the labor market indicators and the number of audits.

|  | (1) | (2) | (3) | (4) | (5) | (6) |
|---|---|---|---|---|---|---|
| VARIABLES | emp | emp | u | u | debt | debt |
| control | 0.00234 ** | 0.00485 *** |  | $(5.61 \times 10^{-5})$ | 2.614 ** | −2.395 |
|  | (0.000928) | (0.00188) |  | $(6.02 \times 10^{-5})$ | (1.089) | (2.411) |
| sq_CONTROL |  | $-1.47 \times 10^{-7}$ |  | $-5.72 \times 10^{-10}$ |  | 0.000304 ** |
|  |  | $(9.61 \times 10^{-8})$ |  | $(3.12 \times 10^{-9})$ |  | (0.000130) |
| lncontrol |  |  | 0.0746 |  |  |  |
|  |  |  | (0.108) |  |  |  |
| Constant | 855.4 *** | 849.9 *** | 5.803 *** | 6.175 *** | 29,391 *** | 39,857 *** |
|  | (84.64) | (84.11) | (0.930) | (0.411) | (8198) | (9369) |
| Observations | 255 | 255 | 255 | 255 | 255 | 255 |
| Number of region | 85 | 85 | 85 | 85 | 85 | 85 |

Standard errors in parentheses *** $p < 0.01$, ** $p < 0.05$.

According to the above calculations, the used data do not allow us to conclude that there is a significant relationship between labor market indicators and the number of audits.

The final stage of the empirical study is the analysis of the influence of the administrative control level on the population income in the region (Table 5).

**Table 5.** Dependence model evaluation results between the level of administrative control and income of the population in the region.

|  | **(1)** | **(2)** |
|---|---|---|
| VARIABLES | income | income |
| control | −0.123 * | −0.118 |
|  | (0.0706) | (0.145) |
| sq_CONTROL |  | $-2.52 \times 10^{-7}$ |
|  |  | $(7.46 \times 10^{-6})$ |
| Constant | 29,459 *** | 29,447 *** |
|  | (1408) | (1420) |
| Observations | 255 | 255 |
| Number of region | 85 | 85 |

Standard errors in parentheses *** $p < 0.01$, * $p < 0.1$.

## 5. Discussion and Conclusions

The table shows that the obtained results do not allow us to draw a meaningful conclusion that an increase in the number of audits leads to an increase in the well-being of citizens.

From a formal point of view, testing of the set hypotheses led to the following results.

*Hypothesis I.* The level of administrative control has a significant (based on the Hausman test, the random effects model is preferable to the fixed effects model) positive effect on the net financial result of organizations in the regions of the Russian Federation. As an economic justification for this fact, we can present an example of the withdrawal of enterprises from the gray zone through administrative audits. This suggests that, despite the direct financial costs associated with tightening control (Blanc 2012), the cumulative effect on the profitability of enterprises is positive.

*Hypothesis II.* This hypothesis is confirmed by real data. In the absence of a proper level of control, an increase in monitoring and supervisory actions leads to an increase in the number of profitable enterprises. However, upon reaching a certain level, further growth of regulation has a significant negative effect on the activities of enterprises in terms of their profitability.

Specialists offer different ways to overcome administrative barriers.

According to other authors, the reduction of administrative barriers in licensing is facilitated by the transition to an electronic form of licensing procedures, and the posting of full information on the official websites of government bodies (Vasiliev et al. 2016).

The researchers speak of codification and systematization of legislation governing administrative procedures as one of the main ways to reduce administrative barriers to business (Shestoperov and Kalinina 2005).

The reduction of administrative barriers in general and in passing state control is associated with general trends in the regulation of economic and other spheres. The level of regulation should reflect the middle ground between two processes in the impact of the state on business: increased demands on certain economic activities and economic deregulation (Guasch and Hahn 1997; Hahn 1998).

It is impossible to completely abandon state control as a kind of administrative barrier; this is due to various types of security. The analysis we conducted recognized the presence

of a certain level of administrative influence in the form of control, which not only does not interfere, but also stimulates the economic development of the regions.

Scientists consider the following ways of optimizing the regulatory impact on various areas of relations: (1) introduction of collaborative and inter-organizational (interdepartmental) governance (co-regulation) (Dung et al. 2023); (2) elimination of duplication of control functions of various administrative bodies; (3) combination of flexibility, efficiency, and consistency; (4) risk-based approach; (5) use of information systems; (6) transparency of mandatory requirements (Blanc 2012). We believe that, in certain cases, the methods of optimizing state control and the ways of minimizing administrative barriers coincide.

The ways we propose to reduce administrative barriers are derived both from the analysis of the proposals of the above authors and according to the results of our study. They are associated with procedural simplification of control and with "saving" administrative impact.

An administrative procedure, its part, or a condition within the framework of any interaction of citizens and organizations with public authorities (federal and regional government bodies) can be considered an administrative barrier. For a business representative and other non-powerful subject, any requirement of power acts as a barrier, while from the point of view of state and other public interests (security interests, interests of consumers of goods and services, etc.), there is a necessary (optimal) level of managerial or regulatory impact on a certain area, which stimulates the development of this sphere in the direction set by the authorities and society. Exceeding this level entails a decrease in economic activity, independence, and investment attractiveness of the sphere. A decrease in this level poses security threats and other risks. The conducted research makes it possible to outline the conditional limits of administrative impact (on the example of control and supervision), which allow making the presence of the state in a certain area as inconspicuous as possible but effective.

Although the analysis of existing studies on administrative barriers has shown that scientists position various conditions and factors (e.g., administrative procedures, their conditions and stages, administrative formalities, regulatory framework, mandatory requirements) as such barriers, they all agree that barriers affect the most important indicators of socio-economic development.

The three approaches to studying administrative barriers that we have identified above (institutional (Auzan and Kryuchkova 2001, 2002; Malikov 2003; Nikolaev and Shulga 2003), integration (Balandina 2011), and procedural (Rodrik et al. 2004; Naryshkin and Khabrieva 2006; Madsen 2009; Mason and Brown 2011; Agamagomedova 2017) offer different understandings of both the barriers themselves and the ways of measuring, evaluating, and calculating their level and degree of impact on the environment and its participants.

Representatives of institutional economics suggest comparing the costs of an economic entity for going through administrative procedures with its private and social benefits, given the income effect (Auzan and Kryuchkova 2002).

Other scientists identify administrative barriers to doing business of an economic nature; they understand various transaction costs that are the reason for the low economic activity of business entities (Gamidullaeva and Tolstykh 2017; Vasin et al. 2018; Gamidullaeva 2019a).

There are isolated studies of the impact of administrative barriers on business on the example of particular regions; besides, the main method is a sociological survey (e.g., of the Orenburg region) (Narovlyanskaya and Kartasheva 2012).

In addition, a study of the relationship between administrative barriers to business and propensity to do business in EU countries is present in modern science (Anders 2011).

The uniqueness of our study lies in the following facts:

- We considered control and supervisory procedures as administrative barriers. Despite the variety of forms of barriers, control and supervision are the most indicative since, in most cases, business faces barriers when passing state control and supervision;

- Controls (inspections, checks) are considered a barrier problem for business, but without an empirical basis (Blanc 2012). The author rightly singles out time, financial, and procedural costs as barriers, and indicates their impact on investment decisions and expectations of potential growth; however, the author does not offer constructions of their connection with economic indicators (Blanc 2012);
- The use of the panel data analysis method makes it possible to take into account both the spatial and temporal components of the considered dependence. Panel data allows us to control for variables that are unobservable and for variables that change over time but differ across regions. Such data make it possible to more accurately take into account the individual heterogeneity of the sample;
- In contrast to some works (Solodilova et al. 2016), where the impact of existing parameters of administrative regulation of entrepreneurial activity on the investment climate in the region is noted, the results of our study are more detailed as they show the impact of the number of inspections on the growth of citizens' well-being. We believe that, in this case, it is appropriate to speak of a complex balance between immediate needs for "security", including in response to "fears" and incidents, and the actual impact on security caused by economic prosperity (Blanc 2012).

Our research results are consistent with other previous studies, for example, with the findings of Dobrolyubova and Yuzhakov (2017) that inspections lead to a decrease in business profitability and a decrease in profits. Additionally, our findings are consistent with the position of Solodilova et al. (2016) that the level of administrative control has a significant impact on the level of investment activity in the region. In the latter case, our results are more complete and revealing, given that the model we have built indicates that there is a certain level of control, and after it is achieved, further strengthening of administrative barriers negatively affects the level of investment in the region. A similar level was not singled out in similar studies, which allows us to speak about a certain novelty of the proposed study.

We identified the following ways to overcome administrative barriers to business when passing state control and supervision:

- Digitization of state control and supervision;
- Transfer of the controlled sphere part to self-regulation;
- Preliminary examination of draft regulatory legal acts;
- Division into categories of individuals to make state control more selective and targeted.

In Russia, the law on administrative procedures has not been adopted yet. There have been several attempts to develop and adopt such an act. The bill of the Federal Law On Administrative Procedures is currently being discussed.

It should be noted that the category of the barrier-free environment is closely linked with such categories as business climate or business environment at a certain (regional, national) level. In this case, the barrier-free administrative environment acts as an ideal model, a vector for the development of administrative regulation, and involves the most favorable business environment or an environment with the most comfortable and favorable business climate.

This paper creates the basis for future investigations in terms of conceptualization and theoretical justification of the impact of various administrative measures on regional business efficiency.

In further research, we plan to carry out the proposed analysis on the empirical data related to other types of barriers. Moreover, we suggest considering the time factor and emphasizing the need to study it not in statics, but taking into account its dynamism.

**Author Contributions:** Conceptualization, L.G.; Methodology, L.G. and S.A.; Writing—original draft, L.G. and S.A.; Supervision, L.G.; Writing—review & editing, L.G. and S.A.; Funding acquisition, L.G. All authors have read and agreed to the published version of the manuscript.

**Funding:** This study was supported by the grant of the President of the Russian Federation for the young Russian scientists' state support on scientific research «Balanced development of the territory based on industrial clusters in the context of theory of "smart specialization"» (grant number: MD-1823.2022.2).

**Institutional Review Board Statement:** Not applicable.

**Informed Consent Statement:** Not applicable.

**Data Availability Statement:** Not applicable.

**Conflicts of Interest:** The authors declare no conflict of interest.

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
