# Peer review of "How Administrative Regulation Institutional Factors Affect the Business Efficiency in a Region: A Case Study of Russian Regions"

_economies, doi:10.3390/economies11030100_

Round 1
Reviewer 1 Report
· Summary
The title of the manuscript indicates that the subject under consideration is how business efficiency in a region is influenced by administrative regulatory bodies, which is a subject manifestly too vast to be covered by a single study. Fortunately, the abstract reveals that this is a case study that concerns some regions of the Russian Federation at a certain point in time. Using a panel data methodology, the authors conclude that “The level of administrative control has a significant […] positive effect on […] (profit minus loss) of organizations in the regions of the Russian Federation.” (Hypothesis I) and that, in terms of profitability of those organizations, there is an optimal level of administrative control (Hypothesis II).
· Comments and recommendations
Let us start with the smaller issues, which are obviously less problematic and, in a sense, also more comprehensible.
Firstly, it is known that when many references are used throughout the text, there is a tendency for slips in the form of their citation. In any case, it is the authors' obligation to prevent the manuscript from being submitted without these slips occurring. To be clearer, I will list below some of the references found so that the authors can see how they are different: “(Gamidullaeva et al. 2020)”; “Nikolaev and Shulga, 2003”; “M. Boeheim et al. (2006)”; “(Fariza 2012, Diaz, 2017)”; “(Pilvere et al. 2013; Rauch et al. 2013; Prykhodko 2015)”; “(Solodilova, et al., 2016)”.
Secondly, authors should always check the version of the file that will be submitted. In fact, it is incomprehensible that, in this case, figure 1 begins on a page (pg. 6) and ends on another (pg. 7).
Let us continue with the more problematic issues.
Authors start their manuscript with something that, in my opinion, is common knowledge, i.e., that the context in which economic agents carry out their activity is relevant to their performance. To be clearer, allow me to quote the second and third sentences of the introductory section: “Different business environments may have differential impacts. The improvement in the institutional environment and increased certainty about the future have much impact on entrepreneurial behavior (Gamidullaeva et al. 2020).”.
As this aspect is a crucial issue for the manuscript, the methodology used to study it should, obviously, take into account that context or environment. In my opinion, the one used does not allow it, much less considering the lack of control variables (please do not confuse with the designation used in the manuscript for the only explanatory variable, i.e., “control”). I really believe that the appropriate methodology would have been spatial econometrics on panel data, as it takes due account of the nature of the data, in which, obviously, spatial location, as an essential element in that context or environment, cannot be ignored.
To allow the authors to move forward, I recommend reading chapters 7 and 11 of the Handbook of Spatial Analysis (which is freely available at https://www.insee.fr/en/information/3635545; accessed November 24, 2022). Obviously, I have no association with this reference.
Reviewer 2 Report
Dear Authors.
I have been keen on reading your article on very interesting topic.
After reading it, I was however, quite confused and do not know what are your resuts based on as well as the recommendations.
Bleow you will find the notes I have to your article:
1) if the article is focused on the Russian market, the theoretical background concerning administrative barriers should be based on the knowledge not only Russian, but also other authors as well. In the Introduction part as well as the Methodology section there are around 90 % of sources of Russian authors . At page 5 there are "a list" of foreign authors, but without any information what they conclude. Please provide wider (does not have to be longer) overview about the problem.
2) I found the article (first 5 pages) focused a lot on categorizatin of barriers and different types of these. Some parts are also overlapping.
3) The Methodology part contains no methodology of the research. It just continues in the Introduction part as a literature overwiev. The are no goals, hypotheses or research questions mentioned.
4) Hypotheses should not be part of results.
5) I do not understand figure 1 as it is on two pages and is not clear which arrow belong to which frame.
6) The data should be part of Methodology section, not Results.
7) The "Discussion" means you should compare your results with other authors. Your results should be stated in the Result part.
8) What was the type of data you work with? Why have youchosen panel analysis? It is not clear out of the description if the panel method is suitable.
9) Please decribe clearly what are your dependent and indepentend variables for each of the model. The variables you are describing are not consistent with the hypotheses.
10) What do you mean by "optimal" or "sub-optimal" level of investment climate? How you define "optimum"? (page 10, lines 447-450)
11) What do you mean by "balanced financial results"? How you can deduct loss from profit??? Which model is the base for this result?
12) How have you identified the ways to overcome barriers? How are these "ways" derived from your results?
13) Sources are not in alphabetical order.
14) What do you mean by the source "President of the Russian Federation" (why there is no name of the president)?
Reviewer 3 Report
The introduction can be inproved to increase the understanding of the relevanse of the study. However, I would suggest that the analysis and conclusions are further developed, especially the theoretical contribution that is said to "...create(s) the basis for future investigations".
It is my opinion that the article is interesting and makes an important contribution, but in my view, the analysis and conclusions are too unclear and need to be structured in a clearer way.
Reviewer 4 Report
- The paper must be correctly structured. Thus, section 2 entitled "Methodology" actually contains theoretical elements. It is only in the section entitled "Results" that elements related to the methodology used can be found.
- Figure 1 must be redone in order to visualize all the elements taken into account to carry out the study.
- The research in the article is mainly based on the papers of Russian researchers, which is partly justified by the fact that the study refers to Russia, but there are representative research, on the topic, of other researchers that are completely neglected in the whole article. Therefore, the paper must include references to representative articles in the field.
Round 2
Reviewer 1 Report
Experience tells me that, in general terms, there are two attitudes on the part of reviewers and authors in their response to reviewers' recommendations: one of a constructive nature and the other of an arrogant and offensive nature. As a reviewer, I try as much as possible to recommend changes in a constructive way. This is how I proceeded when I pointed out that the title of the manuscript was too generic. That's why I found it surprising that the authors, in an arrogant way, responded that "There are no suggestions to change the title of the paper, so it remained the same." I also found it particularly arrogant that the authors did not acknowledge that they did not know how to take into account the methodological recommendation that I constructively placed in point 4 of my first review report.
Reviewer 2 Report
Dear Authors,
you have done a lot of improvements of your manuscript.
The one thing I have to insist on is the decution of loos from profit. It is nonsence to deduct these two variables. Check this in your article and explain what you really wanted to calculate.
